# Current Status of Vaccines for Porcine Reproductive and Respiratory Syndrome: Interferon Response, Immunological Overview, and Future Prospects

**DOI:** 10.3390/vaccines12060606

**Published:** 2024-06-01

**Authors:** Jiuyi Li, Laura C. Miller, Yongming Sang

**Affiliations:** 1Department of Food and Animal Sciences, College of Agriculture, Tennessee State University, 3500 John A Merritt Blvd, Nashville, TN 37209, USA; jli4@tnstate.edu; 2Diagnostic Medicine and Pathobiology, College of Veterinary Medicine, Kansas State University, 1800 Denison Ave, Manhattan, KS 66506, USA; lauramiller@ksu.edu

**Keywords:** porcine reproductive and respiratory syndrome, PRRSV, interferon, vaccine, antiviral immunity

## Abstract

Porcine reproductive and respiratory syndrome (PRRS) remains a formidable challenge for the global pig industry. Caused by PRRS virus (PRRSV), this disease primarily affects porcine reproductive and respiratory systems, undermining effective host interferon and other immune responses, resulting in vaccine ineffectiveness. In the absence of specific antiviral treatments for PRRSV, vaccines play a crucial role in managing the disease. The current market features a range of vaccine technologies, including live, inactivated, subunit, DNA, and vector vaccines, but only modified live virus (MLV) and killed virus (KV) vaccines are commercially available for PRRS control. Live vaccines are promoted for their enhanced protective effectiveness, although their ability to provide cross-protection is modest. On the other hand, inactivated vaccines are emphasized for their safety profile but are limited in their protective efficacy. This review updates the current knowledge on PRRS vaccines’ interactions with the host interferon system, and other immunological aspects, to assess their current status and evaluate advents in PRRSV vaccine development. It presents the strengths and weaknesses of both live attenuated and inactivated vaccines in the prevention and management of PRRS, aiming to inspire the development of innovative strategies and technologies for the next generation of PRRS vaccines.

## 1. Introduction

The swine arterivirus, a main etiological virus causing porcine reproductive and respiratory syndrome (PRRS), is also known as PRRS virus (PRRSV). This infectious disease was first reported in the United States in 1987 and has since been widely detected across the globe [1]. Today, PRRSV has emerged as one of the most important pathogens, exerting substantial economic toll on the swine industry worldwide [2,3]. PRRSV can infect pigs of any age, but the clinical manifestations are particularly severe in pregnant sows, especially during the last trimester, as well in young piglets [4]. Infected pregnant sows may prematurely farrow, resulting in stillborn, partially autolyzed, or mummified fetuses. Young piglets exhibit symptoms such as fever, severe dyspnea, anorexia, lethargy, eyelid edema, and a notable blue or red discoloration of the ears or hindquarters. While finishing pigs, boars, and unbred gilts often present less severe signs, the presence of additional infections in PRRSV-infected pigs can exacerbate and complicate the situation, potentially leading to more severe reproductive failure in pregnant sows or to a respiratory disease complex in young piglets. Therefore, the prevention of PRRS and related diseases is among the top priorities of swine producers [5,6].

PRRSV belongs to the enveloped RNA viruses of the *Arteriviridae* family, which is part of the *Nidovirales* order together with *Coronaviridae*, the family that includes the virus that caused the recent human COVID-19 pandemic [7]. PRRSV has a single-strand positive RNA genome of about 15 kb [(ss(+) RNA)] containing at least 11 open reading frames (ORFs) to encode viral structural and non-structural proteins (Nsps) [8]. These 11 ORFs include the classical ORF1-10 and a recently identified ORF embedded within the ORF1a region, which is expressed by a programmed -2 ribosomal frameshift (-2 PRF) of the Nsp2 ORF, resulting in a longer Nsp2TF that determines a trans-frame (TF) fusion with the N-terminal two-thirds of Nsp2 [9]. Approximately 75% of the PRRSV genome consists of ORF1a and ORF1b at the 5’-terminus, which encode two viral polyproteins (pp1a and pp1ab) that are further processed through autocatalysis into Nsps to mediate viral replication and pathogenesis in the host [10]. The remaining 25% of the viral genome at the 3’-terminus contains eight ORFs encoding viral structural proteins, including the envelope protein (E), glycoprotein 2 (GP2) to GP5, the membrane protein (M), and the nucleocapsid protein (N) [11]. Owing to the inherent absence of proofreading capability in the RNA-dependent RNA polymerase (RdRp) of PRRSV, the viral genome exhibits a high propensity for mutation and recombination [12]. Notably, most of the alterations predominantly occur within ORF5, a viral gene characterized by high variability. This 603-nucleotide gene encodes the major envelope protein GP5, which possesses several immune epitopes [13]. Sequence profiling of the ORF5 region of PRRSV field strains has emerged as a prevalent approach for genotype and strain differentiation, as well as for comprehending the viral diversity per molecular and pathogenic evolution. For instance, several subtypes, such as NVSL-97-7895, KS-2006-72109, and NADC34, have been identified in the United States. Additionally, highly pathogenic PRRSV (HP-PRRSV), recombinant PRRSV-1 TZJ2134, and the outbreak strains NADC30/NADC34 have been recognized in Asia [14,15,16,17]. These outbreaks have led to significant economic losses in global pig farming. Relevant studies have reported an alarming mortality rate of greater than 50% caused by these emerging pathogenic strains in newborn pigs in several countries, including Austria, Italy, Belarus, and China [18]. In our review, we systematically identified relevant references through comprehensive searches in PubMed, Scopus, and Web of Science using keywords like “PRRSV”, “KV vaccines”, “MLV vaccines”, “Immunology”, and “Advantages and Disadvantages”. We focused on the efficacy, safety, and immunological aspects of KV and MLV vaccines for PRRSV. Titles and abstracts were screened to filter out irrelevant studies, and full-text articles were reviewed to ensure they met the inclusion criteria. Additionally, we examined the reference lists of the selected articles to identify further relevant studies.

## 2. PRRS Vaccine Immunology: An Overview

A vaccine is a biological preparation designed to confer immunity against a specific disease by prompting the production of antibodies, fostering cellular immunity, or both. This preparation may be derived from the pathogen causing the disease, its by-products, or a synthetic analog modified to serve as a specific antigen without causing the disease itself [19,20]. One of the earliest vaccines for pigs, targeting the bacterium *Erysipelothrix rhusiopathiae*, was developed by Louis Pasteur in France in 1883 [21]. Following this pioneering work, vaccination strategies expanded to protect swine against various pathogenic threats. With the industrialization of swine production and the advent of intensive farming practices, vaccination has become an indispensable and critical component of swine production. The evolution of swine vaccines has predominantly utilized either attenuated live or inactivated forms. The underlying considerations not only emphasize the consistency and safety of the vaccine product but also facilitate its scalability for commercial production and wider distribution. Today, the role of vaccines in swine health management remains paramount, underpinning the ongoing efforts to safeguard the well-being and productivity of swine populations globally [20].

The progression of PRRSV infection involves three stages: an initial acute phase, a subsequent persistent phase, and, finally, extinction if effective immune clearance is stimulated [22]. The efficacy of modified live virus (MLV) vaccines in swine is attributed to their capacity to elicit both cell-mediated and antibody-mediated immune responses, notably enhancing mucosal immunity [23]. MLV vaccines utilize an attenuated form of the virus, which, while capable of stimulating a robust immune reaction, generally does not lead to disease manifestation or results in only a mild clinical presentation [24]. During the initial phase of infection, various factors such as infection, inflammation, stress, physical injury, or tissue damage may perturb the homeostatic equilibrium within a swine population [25]. This nuanced observation underscores the complex interplay between vaccine-induced immunity and the maintenance of health and disease resistance in swine herds. In pigs, evolutionary processes have led to the formation of two interlinked defense systems against microbial pathogens: the innate and adaptive immune systems [26].

The innate immune system, acting as the initial line of defense, exploits a collection of germline-encoded receptors known as pattern recognition receptors (PRRs) to detect and launch responses against pathogens [27]. These PRRs possess the capability to perceive molecular motifs conserved within each class of infectious agents, known as pathogen-associated molecular patterns (PAMPs), essential for the pathogen’s lifecycle and proliferation [28]. Such PAMPs, including viral dsRNA, bacterial lipopolysaccharides, peptidoglycans, and flagellin, are exclusive to microbes but absent in the host, thereby enabling the host’s immune system to differentiate between self-entities and non-self-entities at the molecular level [22]. These receptors, widely expressed on the surface and in the interior of various cells, particularly immune cells—including macrophages/monocytes, dendritic cells, natural killer (NK) cells, mast cells, neutrophils, and eosinophils—play a crucial role in the immune response. The recognition of certain PAMPs by cognate PRRs activates the immune response, leading to the production of a series of innate immune effectors, including antimicrobial peptides, inflammatory cytokines, and interferons (IFNs), which mediate antiviral responses particularly against intracellular pathogens such as viruses. In an antiviral response, for instance, IFNs, especially type I and III IFNs, mediate the expression of hundreds of IFN-stimulated genes (ISGs), which may directly restrict or inactivate viruses and further bolster the body’s adaptive response against viral infections [22,26,29].

The adaptive immune system’s efficacy hinges on the development of a varied array of antigen receptors present on T- and B-lymphocytes, followed by the activation and clonal proliferation of cells equipped with specific antigen receptors. The adaptive immune response includes both T-cell-mediated cellular immunity (CMI) and humoral immunity (HI), the latter of which involves the production of specific antibodies by effector B-cells. The initiation of adaptive immunity is contingent not merely on the direct recognition of antigens by these receptors but also on critical signals emanating from the innate immune system [22,30].

For instance, type I IFNs (IFN-α/β are the subtypes most studied) exert a direct regulatory influence on T-cells via interferon-α/β receptors (IFNARs) located on the T-cell surface. This interaction influences T-cells’ survival, proliferation, secretion of IFN-γ, and cytotoxic activity, and the development of memory T-cells [22]. Differentiation between αβ and γδ T-cells is determined according to the distinct gene sets expressed to form their T-cell receptors (TCRs). Within αβ T-cells, TCR coreceptors delineate two primary subsets: CD4+ and CD8+ T-cells, each serving distinct roles within the immune system [31]. CD4+ T-cells enhance the immune response by aiding B-cells in antibody production and exerting antiviral functions [32]. On the other hand, the clonal expansion of CD8+ T-cells targets and destroys virus-infected cells, thereby increasing the immune system’s cytotoxic capabilities [33]. A study of PRRSV infection revealed that the depletion of CD8+ T-cells during the early stages did not exacerbate disease severity nor affect the pigs’ capacity to eliminate the virus [34]. Conversely, a CTL phenotype was characterized by high levels of both CD4+ and CD8+ (CD4+CD8+high), which demonstrated the ability to destroy infected cells within 3–6 h post infection. Another CTL phenotype (CD4+CD8-) exhibited cytotoxicity towards infected cells at a later phase, 16–24 h after infection. These observations suggest that both CD4+ and CD8+ T-cells can independently mount effective responses; however, their combined action is synergistically more potent, highlighting the complex and dynamic nature of T-cell-mediated immunity in confronting viral infections [35]. For γδ T-cells, this group shares several features with αβ T-cells, such as strong cytotoxic and regulatory functions, including the ability to induce dendritic cell maturation, cytokine production, and the maintenance of immunological memory [36]. They respond rapidly to infections, especially in mucosal immunity, and have the capability to present antigens and recognize a broad spectrum of antigens directly through their TCRs, without the need for MHC co-signaling. This capability positions γδ T-cells as a bridge between the innate and adaptive immune systems, categorizing them as unconventional T-cells due to their distinct biological characteristics and pivotal role in the immune response [31,36,37]. Recent research has further illuminated the role of γδ T-cells, in their collaborative effect with CD4+ cells in safeguarding against transplacental infections. This was particularly observed in a quantitative study involving PRRSV-infected sows [38].

Following the activation of B-cells by T-cells, which leads to the production of specific antibodies by effector B-cells, numerous studies have indicated that the infection of PRRSV predominantly triggers a specific humoral immune response [22,39,40,41]. This response is likely attributed to a sustained humoral reaction resulting from the continuous presence of the virus in the tonsils and lymphoid organs [42,43,44]. Just as type I IFNs are critical in activating T-cells, they similarly play a fundamental role in the early stages of B-cell activation, substantially affecting the antibody response mediated by B-cells [43,45]. The generation of circulating neutralizing antibodies (NAbs) facilitated by this interaction is instrumental in the efficient eradication of the virus [46]. Moreover, type I IFNs contribute to various aspects of B-cell function, including their activation, antibody secretion, and isotype switching during viral infections, underscoring their comprehensive influence on the immune response to pathogens [47,48]. Protective humoral immunity, established after vaccination or pathogen exposure, hinges on two main defenses: the immediate secretion of antibodies by plasma cells and the long-term surveillance by memory B-cells [49]. Memory B-cells, upon re-encountering the antigen, rapidly proliferate and mature into plasma cells that produce high-affinity, isotype-switched antibodies, swiftly enhancing the antibody titers [50]. This dual approach ensures both immediate and sustained immune protection, with memory B-cells playing a pivotal role in bolstering the body’s rapid response to re-infection [51]. In swine, studies have shown that an early antibody response to PRRSV infection is observable by the fifth day post infection [46,52]. Initially, these antibodies, targeting the virus’s N proteins and M proteins, lack the ability to neutralize the virus [53]. It was once thought that PRRSV-neutralizing antibodies did not emerge until after four weeks. However, more recent findings have refined this timeline, indicating that neutralizing antibodies can start appearing between 7 and 10 days post infection [54]. Despite their earlier appearance, the concentration of these neutralizing antibodies typically remains low, with considerable variability being observed in the potency and persistence of the neutralizing response across different swine populations [22,54]. This delayed and weak neutralizing antibody response might be attributed to several factors, including the infection’s multiphasic nature, an ineffective initial innate immune response, and the slow maturation and proliferation of cell-mediated immune responses [22]. In contrast, one study [55] highlighted that piglets with higher levels of maternal-transferred NAbs exhibited delayed and reduced viral loads in their sera and improved survival rates against wildtype PRRSV infections compared to fattening pigs. These piglets maintained relatively high NAb titers for at least four weeks. Thus, suckling piglets with elevated maternal NAb levels demonstrate enhanced immune protection, effectively countering PRRSV during the early nursery phase. Further investigations revealed that while NAbs contribute to protective immunity, their effectiveness is often limited for specific PRRSV strains, underscoring the strain-specific nature of most NAbs [56]. Nonetheless, some pigs develop broadly reactive NAbs that can neutralize a diverse array of heterologous PRRSV isolates. This capacity for cross-neutralization suggests that such antibodies can play a vital role in the immune defense against PRRSV, influencing the effectiveness of vaccines and contributing to immune memory [56,57].

To counter viral infections, hosts have developed intricate and advanced immune systems. However, PRRSV has crafted evasion techniques to bypass these defenses, thus facilitating its proliferation. This involves the manipulation of cellular immune signaling pathways, notably through the suppression of IFN responses. Viruses employ various tactics to counteract the IFN system, such as inhibiting IFN production, disrupting IFN-mediated signaling pathways, and hindering the function of IFN-induced antiviral proteins. Specific examples of PRRSV’s immune evasion include the suppression of IFN-β production by the N protein, which it accomplishes by inhibiting TRIM25 expression and its mediated ubiquitination of RIG-I [58]. Additionally, PRRSV’s non-structural proteins nsp1α/β prevent the dsRNA-induced activation of interferon regulatory factor 3 (IRF-3) [59], while Nsp11 collaborates with a swine-specific deubiquitinase to remove linear ubiquitin chains from the NF-κB essential modulator, thereby inhibiting NF-κB signaling [60]. Furthermore, in a nuanced interplay of micro-RNA-mediated antiviral mechanisms within pigs, PRRSV infection elevates the levels of a specific host micro-RNA, miR-373. This elevation aids in the virus’s replication by impeding IFN-β production, targeting IRAK1, IRAK4, and IRF4—key players in the interferon response pathway [61,62,63].

From an immunological perspective, a vaccine that enhances the host’s defense against viruses by boosting innate immunity, as well as the functions of CD4+ T-cells, CTLs, and B-cells, would be highly beneficial [22]. PRRS is characterized by weakened innate immunity, particularly in the production of type I IFNs, and a compromised, sluggish adaptive immune response. Given the critical role of type I IFNs in the growth and development of adaptive immune mechanisms, there is a highlighted need for targeted vaccine development to overcome PRRSV’s evasion strategies.

## 3. Interferon Response in PRRS Vaccine Studies

Interferons, as a group of immune cytokines, are renowned for enhancing antiviral defenses in high vertebrates. The ancestral genes of IFNs were first identified in jawed fish, coinciding closely with the emergence of adaptive immunity in animals [64,65]. Three types of interferons—types I, II, and III—have been discovered and functionally studied regarding antiviral reactions and vaccine designs in animals and humans [66,67]. For example, type I IFNs include several subtypes of IFN-α/-β/-ε/-κ/-ω that commonly exist in all studied mammalian species, as well as other species-specific subtypes such as IFN-δ/-τ/-ζ, all of which bind to a common cell-surface IFN-α/-β receptor (IFNAR) consisting of two subunits of IFNAR1 and IFNAR2 [66,68]. Type II IFNs, represented solely by IFN-γ, are primarily produced by NK and T-cells, and mediate immune signaling through a distinct IFN-γ receptor (IFNGR1/2) expressed on various immune cells [69]. Type III IFNs, also known as IFN-λ, primarily act on mucosal epithelial cells through cognate IFN-λ receptors consisting of two subunits of IFNLR1/IL10R2 [70]. The expression of IFN genes, particularly of type I and III IFNs, is regulated through several signaling pathways, including Toll-like receptor (TLR) signaling and retinoic acid-inducible gene I (RIG-I)/MDA5 pathways, in order to recognize virus-derived PAMPs and further activate responsive IFN production [71]. IFN ligands, in turn binding to respective cognate receptors on target cells, trigger the expression of a series of genes known as IFN-stimulated or IFN-induced genes (ISGs), which exert diverse functionality in antiviral, antiproliferative, and various other immunomodulatory roles [71]. Because of the critical role of IFNs in antiviral regulation, viruses, especially those causing successful infections and impactful diseases in animals, have co-evolved various mechanisms to counter against the host IFN system, which have been extensively reviewed elsewhere [71,72]. In brief, these viral IFN-antagonistic mechanisms may target different stages of IFN production and action processes in independent or synergistic ways, which include (1) inhibiting effective IFN gene expression, and thus IFN cytokine production, via targeting the main components of upstream PAMP-activated signaling pathways that lead to IFN gene expression; (2) suppressing IFN action signaling post IFN–receptor interaction and intracellular JAK-STAT signaling pathways in IFN-responsive cells to repress ISG expression; (3) directly inactivating the antiviral activity of certain ISGs; (4) indirectly interfering with IFN cytokine production or action through targeting miRNA or other epigenetic regulators; and (5) indirectly further counteracting IFN antiviral signaling via cross-talk with immune-suppressive signaling pathways such as that mediated by the suppressor of cytokine signaling 3 (SOCS3). It is noteworthy that as viruses may evolve multiple antagonistic proteins to interfere with individual components in IFN signaling pathways, some viral proteins, especially those derived from non-structural proteins of notorious RNA viruses, including SARS-COV2 and PRRSV, may be capable of targeting host IFN responses at different steps of the IFN signaling pathways [71,72].

As exemplified in PRRSV and porcine IFN studies, these virus and host IFN systems seem to be undergoing a constant arms-race coevolution [73,74,75,76]. For instance, most pathogenic PRRSV infections suppress IFN expression, particularly measured in IFN-β, through various mechanisms including Nsp1, Nsp4, Nsp9, and structural N proteins, which have been reviewed previously [73,74,75,76]. Recent studies show that PRRSV also interferes with adaptor protein IPS-1 activation in the RIG-I signaling pathway, specifically, it seems, by Nsp4 working through its protease activity, as well as Nsp11 targeting IRF9 via a NendoU activity-independent mechanism and cleaving the mitochondrial antiviral signaling protein (MAVS) in a proteasome- and caspase-independent manner. Additionally, other studies have shown that over-expressing certain receptors like LSM14A, and specific miRNAs such as miR-23, miR-378, miR-505, etc., can upregulate IFN-β activity, enhance its production, and reduce the secretion of related inflammatory cytokines, thereby inhibiting PRRSV infection [77,78]. All of these findings underscore the crucial role of IFNs in restricting viral infection and modulating the effective immune responses underlying viral diseases; therefore, the modulation of effective IFN action provides a promising approach to enhance vaccine-induced immune induction and protection efficacy. For instance, Nan et al. [79] discovered that the PRRSV strain A2MC2 induces IFN production in both MARC-145 and PAM cells, unlike most pathogenic PRRSV strains that suppress IFN induction, suggesting that coordinating IFN stimulation could effectuate vaccines for better protective immunity against PRRS in addition to inducing direct IFN antiviral activity, as shown in another study on oral low-dose IFN-α treatment in pregnant sows [24]. To test the direct role of IFN usage in promoting vaccine effects for PRRS vaccine design, Du et al. developed a DNA vaccine that co-expresses PRRSV’s GP3 and GP5 proteins with porcine IFN-α/γ. The results indicated that this vaccine construct provided both immediate and long-lasting protection against HP-PRRSV challenges in pigs. Compared to the control construct that expressed GP3/GP5 only, the IFN-enhanced GP3/GP5 construct elicited significantly higher PRRSV-specific antibody responses, T-cell proliferation, and levels of IFN-γ and IL-4. Additionally, it resulted in almost no clinical signs, no lung lesions, and significantly lower viremia [80]. The IFN incorporation not only enhanced vaccine performance through synergistic effects but was probably also effective on its own per direct IFN-mediated antiviral stimulation. According to another study by Yu et al., recombinant porcine IFN-α (rPoIFNα) showed immunopotentiating activity when used in conjunction with a PRRSV KV vaccine. Administered at doses of 400,000 U or 4,000,000 U, the rPoIFNα significantly boosted antiviral immune responses, including increases in both Th1 and Th2 responses and sequential titers of neutralizing antibodies against challenging PRRSV in the treated pigs [81].

As has been revealed by genome-wide studies across representative vertebrate species, pigs contain the largest IFN complex, particularly the type I IFNs, which consist of nearly 60 IFN-coding genes that encode 7 IFN subtypes, including multi-gene subtypes of IFN-α, -δ, and -ω, as well as single-gene subtypes of IFN-β, -ε, and -κ. While subtypes such as IFN-α and -β have been widely studied in many species, the unconventional subtypes such as IFN-ω have barely been investigated [63,69]. Our recent research used an infectious DNA clone of the PRRSV-P129 for virus replication-competent expression of a cohort of optimized subtypes of porcine IFNs based on a functional characterization of their antiviral responses. As expected, these MLV-129p-IFN constructs induced comparable or better protection compared with a commercial vaccine in grower pigs regarding the body temperature, lung lesion score, and virus titer, as seen in [82,83] (and unpublished data). Furthermore, these studies also profiled signature gene-responsive pathways using transcriptomic analyses in the livers of pigs vaccinated with MLV-129p-IFNmix to reveal an obtained IFN response mediated by the virus replication-competent expression of IFNs in vivo [82]. Consistent with the multifunctional role of IFNs, we detected 197 significantly differentially expressed genes (DEGs) that were enriched in pathways of inflammation and stress responses, in addition to those spanning both innate and humoral immune responses. In contrast to the upregulation of common ISGs, we also observed a downregulation of interleukin 2 (IL-2) and LAG3 inhibitory signaling, which functions in maintaining T-cell proliferation and activation. These findings suggest that introducing unconventional IFNs in PRRSV MLV vaccines may stimulate the interferon response in a timely manner and lead to better protection in the tested period [82]. We also performed a comparative transcriptomic study regarding small-RNA composition and differential micro-RNA responses to PRRSV infection and treatments with different subtypes of type I IFNs, particularly the IFN-α and IFN-ω subtypes, in porcine alveolar macrophages (PAMs). The small-RNA transcriptomic analysis identified 331 known and 55 novel porcine miRNAs from the PAM samples, accounting for 44% of all miRNAs identified in various tissue and cell types of pigs. The PAMs treated with IFN-ω5 enriched the highest number of miRNAs. A cluster analysis revealed that similar patterns of significantly differentially expressed miRNAs were observed among the treatments with the vaccine PRRSV strain and antiviral IFN-α1/-ω5 subtypes. Gene ontology and pathway analyses of the genes targeted by these miRNAs indicated broad enrichments in metabolic, inflammatory, antitumor, and antiviral pathways, especially those involving macrophage activation and IFN signaling. Given that tissue macrophages, especially PAMs, are primary infection sites of PRRSV and critical for PRRS pathogenesis, our findings underscore extensive gene responses involving both protein-coding and non-coding RNA genes that are associated with effective vaccination, as well as the multifunctional role of IFNs in the immunomodulatory and antiviral regulation underlying vaccine effects [83]. The importance of unconventional IFNs was further demonstrated in animal studies conducted by Miller et al. [63,83,84,85]. Their research involved in vivo experiments with pigs to compare the clinical and immune responses elicited by novel vaccine candidates of MLV-p129-IFN constructs. These vaccine candidates expressed a cohort of optimized antiviral IFNs. Their protection effect was tested in parallel with a commercially available PRRSV vaccine against a challenge with a contemporary PRRSV strain, NADC-34. The results indicated that IFN groups and the commercial MLV group returned to baseline temperature levels by post-challenge Day 9, whereas the Mock/NADC-34 group’s temperature levels remained elevated. While all groups showed a temperature peak at Day 2 post challenge, by Day 14, the IFN groups exhibited significantly lower levels of viral RNA compared to the Mock/NADC-34 group. Over time, the comparison between the IFN groups and the commercial MLV group showed no significant differences, suggesting that the MLV-p129-IFN vaccine candidates were as effective as the commercial vaccine in controlling viral replication. In addition, pigs vaccinated using MLV-p129-IFNs demonstrated a significant immunological advantage after the vaccination and challenge, characterized by a robust Th1 cytokine response, sustained antibody levels, and well-regulated immune activation. This should be correlated to the replication-competent IFN effect in counteracting the PRRSV-induced suppression of interferon signaling, thus leading to an enhanced immune response overall [63,83,84,85].

## 4. MLV and KV Vaccines—Current Scenario and Emerging Research for PRRS Prevention

Since the identification of PRRSV, various technologies have been employed to continuously develop different types of vaccines, such as the subunit vaccine, synthetic peptide vaccine, and adenovirus vectored vaccine, to counter this incessantly mutating virus. However, due to the severe clinical symptoms associated with pathogenic PRRSV strains, only vaccines based on either modified live virus (MLV) or killed virus (KV) vaccines have been proven to be of practical use [86,87,88]. Each of these vaccines has its own set of advantages and disadvantages observed in practical applications (Table 1). Although MLV vaccines generally confer effective protection against homologous viruses and partial immunity to heterologous strains, concerns regarding their safety persist due to the potential for reversion to virulence and recombination between the vaccine and virulent field strains [88,89]. The primary advantage of KV vaccines is their safety profile. So far, there have been no reported adverse effects on porcine health associated with KV vaccines [90]. However, their efficacy is generally perceived to be inferior compared to that of MLV vaccines [86,88].

### 4.1. MLV Vaccines

The Ingelvac PRRS MLV vaccine, introduced by Boehringer Ingelheim and first deployed in North America in 1994, has demonstrated long-tested protection against respiratory diseases and lung lesions in the face of heterologous single and dual challenges from PRRSV-1 and PRRSV-2 [91,92,93]. This has paved the way for the wide application of PRRS MLV vaccines over nearly three decades. Currently, there are five typical commercially available MLV vaccines utilized in the United States, with this number being seven in Europe and four in Asia (Table 2). The geographical variation in PRRSV species distribution and disease severity necessitates the application of distinct types of MLV vaccines across different continents. For instance, in Eastern European countries such as Czechia, Belarus, and Hungary, there are occurrences of highly pathogenic PRRSV-1 subtype strains that lead to severe respiratory disease symptoms [94,95]. Conversely, in Western and Northern European countries, such as Germany, Denmark, and France, PRRSV infections of the classical PRRSV-1 strains primarily result in reproductive disorders in breeding pigs and respiratory malfunctions in growing and fattening pigs [94,96]. The varied disease profiles among different subtypes underscore the need to employ distinct MLV-1 vaccines to effectively manage the spread of PRRSV-1. Recurring instances have been observed for PRRSV-2 too, being initially documented in North America before its dissemination to Asia. Moreover, newly emerging HP-PRRSV strains have surfaced, resulting in substantial economic losses in countries like China, South Korea, and Thailand [97,98,99].

Currently, seven MLV1 vaccines are commercially obtainable and in use across European and Asian countries (Table 2). There are numerous reports corroborating the efficacy of these MLV1 vaccines in mitigating respiratory diseases and natural infections, in the context of both viremia levels and the proportion of viremic animals [100,101,102,103,104,105,106]. The evaluation of vaccines against a virulent PRRSV-1 isolate (strain AUT15-33), which caused a severe clinical outbreak, revealed two promising candidates. Administration of the ReproCyc PRRS EU vaccine to gilts led to a reduction in viremia, fetal damage, and transplacental transmission. Similarly, the Ingelvac PRRSFLEX EU vaccine exhibited its effectiveness by enhancing the average daily gain (ADG), curtailing viral shedding through oral fluids, and alleviating the severity of lung lesions and the viral load in tissue samples under experimental conditions [103,106]. For a vaccine-derived recombinant PRRSV-1 Horsens strain from an outbreak in Denmark, MLV vaccines such as Porcilis PRRS, Unistrain PRRS, and Suvaxyn PRRS demonstrated effective protection [107]. Notably, the group vaccinated with Porcilis PRRS exhibited a higher total viral load post challenge compared to those vaccinated with the other two MLVs. At the amino-acid level, the Unistrain vaccine strain (Amervac) shares 97.3% of its identity with the “Horsens” strain in GP5, whereas the Porcilis (DV) and Suvaxyn (96V198) strains show 91.5% and 91.1% shared identities, respectively. These findings suggest that MLV vaccines with heterologous strains can efficiently control PRRSV outbreaks caused by the “Horsens” strain, though variations in protection levels exist between different vaccines [107]. Another study highlighted the suboptimal performance of the Porcilis PRRS vaccine, observing a partial loss of attenuation in the vaccine strain after just two passages in an experimental setting. Contact pigs exhibited more pronounced hyperthermia, reduced average daily weight gain (ADWG), and heightened levels of PRRSV viremia and nasal shedding compared to the vaccinated pigs. While the authors noted that such occurrences were rare, the study suggested that the accumulation of mutations through multiple passages in pigs could be a genetic factor leading to the reduced attenuation and potential reversion to virulence in vaccine strains [108]. It is also important to note that for several herds affected by the Horsens strain outbreak in 2019, the outbreak was effectively contained through mass-vaccination strategies, prominently using the Porcilis PRRS vaccine [109]. While a MLV vaccine generally demonstrates effectiveness, it shows limited efficacy against mutated PRRSV variants. Particularly for PRRSV-1 subtype 3 strains like the highly lethal “Lena” variant, vaccines such as Unistrain PRRS and Ingelvac PRRSFLEX EU provide only partial clinical and virological protection [110,111].

There is limited peer-reviewed information available regarding the efficacy of MLV1 vaccines against PRRSV-2. This is largely due to PRRSV-2’s limited pathogenic role in Europe, the restriction on the use of PRRSV-1 MLV in China, and the sporadic nature of PRRSV-1 transmission in North America [95,112]. Although PRRSV-1 MLV vaccines have exhibited some efficacy against PRRSV-2, especially during the initial infection stages, further evidence indicates that the efficacy of MLV1 vaccines may be contingent on the challenge virus strain. Generally speaking, MLV1 vaccines offer only limited protection against PRRSV-2 infections [93,113,114].

PRRSV-2 MLV vaccines (MLV2), evaluated and employed worldwide, exhibit efficacy against respiratory diseases [114,115,116,117]. For example the latest commercial vaccine, Prevacent PRRS, has been shown to elicit varying degrees of efficacy and immunogenicity against four heterologous strains of PRRSV-2, which are phylogenetically distant [114]. Significantly, despite being in use for over 20 years, Ingelvac PRRS MLV continues to effectively protect against recently identified recombinant PRRSV-2 strains in China and South Korea [91,118]. Recent studies have highlighted the effectiveness of the Ingelvac PRRS MLV vaccine in pregnant sows, showing that they confer strong immunity to their piglets. The piglets, benefiting from enhanced maternally transferred neutralizing antibodies, demonstrate delayed and comparatively mild viral loads in their sera. This results in higher overall pig survival rates when exposed to wild PRRSV infections. Notably, the elevated NAb titers in piglets are maintained for at least four weeks postpartum [55]. Moreover, it has been found that pigs vaccinated with the attenuated Chinese HP-PRRSV vaccine (JXA1-R) were resistant to infection by the North American PRRSV strain, NADC-20 [80]. One study conducted in the United States has suggested that the observed efficacy of the vaccine in providing homologous protection for pig herds across different times and geographic regions could be attributed to the potential decrease in within-lineage viral diversity, a result of repeated introductions of vaccine-like viruses through the use of modified live vaccines [119].

While MLV1 vaccination typically does not induce sterilizing immunity sufficient to entirely block the infection of heterologous strains in swine, numerous field studies have suggested that MLV2 can offer partial protection against these strains [120]. This partial protection is manifested in a delayed onset of viremia, fewer days of clinical fever, diminished lung lesion levels, and less severe clinical signs compared to unvaccinated animals. More specifically, MLV2 vaccines have demonstrated significant protection in the event of a PRRSV-1 strain challenge [113,115,121,122]. Examples such as Ingelvac PRRS MLV and Fostera PRRS have demonstrated their efficacy in reducing viremia, lung lesions, and the count of PRRSV-positive cells in response to challenges with either PRRSV-1 or PRRSV-2. This could be attributed to the induction of equal levels of IFNγ-secreting cells (mainly activated T-cells and natural killer cells) against both PRRSV-1 and PRRSV-2 by PRRSV-2 MLV vaccines, an effect not observed with PRRSV-1 MLV vaccines [113,114]. This is applicable in the current scenario, where both PRRSV-1 and PRRSV-2 circulate concurrently in several Asian countries, and no commercially available MLV vaccine covers both these strains, so the application of a single MLV2 vaccine emerges as the preferred clinical solution for producers to control both PRRSV types [95,97,98,119,123]. However, a recent study on a chimeric vaccine that targets two genetically diverse PRRSV2 strains has shown promising results. The vaccinated pregnant sows in the study exhibited no significant viremia post vaccination. Furthermore, piglets from these vaccinated sows displayed reduced serum viral loads and fewer lung lesions compared to those from non-vaccinated sows. Thus, this vaccine was identified as a safe and effective vaccine candidate, providing protection against multiple PRRSV strains. This breakthrough suggests a significant potential for the future development of chimeric PRRSV vaccines, possibly including both PRRSV-1 and PRRSV-2, aiming to surpass the efficacy of existing vaccination strategies [124].

### 4.2. Potential Risks and Challenges of MLV Vaccines

The most salient characteristic of MLV vaccination is its unpredictable efficacy. Given that all PRRS MLVs undergo replication within the host organism—an indicator of their live state—numerous safety concerns have arisen. These include transmission, recombination, and reversion to virulence [125]. Research indicates that pigs vaccinated with MLVs may exhibit viremia for up to four weeks post immunization, thereby potentially enabling the spread of the vaccine virus to unvaccinated animals [86]. Furthermore, PRRS MLV-vaccinated boars have been observed to shed the MLV virus via semen for a comparable duration. The prevalence of artificial insemination in global swine production raises the risk of widespread transmission of swine viral pathogens due to this persistent shedding [126,127]. One outbreak of PRRSV in Switzerland was traced back to imported semen from Germany, underscoring the inherent risks in importing semen from non-PRRS-free countries, even if the studs are deemed negative. Current monitoring protocols in boar studs may not be sufficient for the timely detection of an infection, with sow/herd infections remaining possible even with limited semen doses [128]. Another recent study aimed to evaluate the safety profiles of all modified live virus (MLV) vaccines available commercially in Europe for combating PRRSV using uniform experimental conditions [129]. The results indicate that the four MLV vaccines (Ingelvac PRRS MLV, Porcilis PRRS, Amervac PRRS, Pyrsvac-183) examined are clinically safe, as evidenced by their inability to replicate in porcine alveolar macrophages (PAMs) within lung tissues. However, it was observed that sentinel pigs, which were not vaccinated, contracted the infection without displaying clinical symptoms. This suggests that the transmission of the vaccine viruses from vaccinated to in-contact pigs most likely occurs through the oronasal route. These findings underscore that a potential risk of transmission is inherent to PRRSV vaccine strains, despite their clinical safety and reduced replication in PAMs [129].

The underlying mechanism of PRRSV recombination remains inadequately elucidated. Initial descriptions of homologous recombination within nidoviruses were documented utilizing the mouse hepatitis virus (MHV), a coronavirus variant, where it was estimated that approximately one-quarter of progeny in co-infected cells exhibit recombinant characteristics [130]. Some research posits that the replication process of viral RNA may generate intermediates that are shorter than the complete RNA sequence. These foundational observations have fostered the development of a hypothesis asserting that MHV replication is characterized by discontinuous transcription, a process wholly dependent on the template-switching functionality inherent to the viral RdRP. This discontinuous transcription mechanism, pertinent to the expansive unsegmented genomes of MHVs and analogous coronaviruses, facilitates the generation of subgenomic negative-sense RNAs via a copy-choice mechanism. This involves the RdRP shifting from one RNA template to another during replication. As a result, RNA recombination is proposed to occur simultaneously with RNA replication in this model, with these RNAs subsequently serving as templates for mRNA synthesis [131,132]. This is substantiated by field reports where PRRSV isolates from subsequent outbreaks showed nearly identical nucleotide sequences to those of vaccine strains from various parts of the world [131,133,134,135]. Meanwhile, some instances of recombination may not have detrimental consequences. In the United States, a recombinant virus was isolated from an Indiana farm, identified as a natural recombinant derived from Ingelvac PRRS MLV and Prevacent PRRS through phylogenetic and molecular evolutionary analyses. Despite originating from two commercially available PRRSV-2 MLV vaccines under field conditions, no gross or microscopic lesions suggestive of a PRRSV infection were observed in lung tissues, suggesting that this recombinant was non-virulent in swine [136]. Recombination may, in certain circumstances, trigger reversion to virulence. An example of this was observed in Denmark, where a recombinant virus was isolated from a PRRSV-negative boar station. The virus was identified as a natural recombinant, derived from the Unistrain PRRS and Suvaxyn PRRS vaccine strains. Despite both parental strains being attenuated vaccine strains, this recombinant strain had regained substantial virulence. The resulting infection in growing pigs, as well as pregnant sows, was comparable to or even exceeded those induced by the typical PRRSV-1 subtype 1 strain [107]. The reversion to virulence is influenced by various factors, including the frequency of the vaccine’s use, correct application of the vaccine, the generation chosen to be utilized as a vaccine, and the characteristics of the candidate vaccine strain. Occasionally, these elements can exacerbate the situation. For example, Danish field isolates derived from a vaccine were sequenced and compared with the parental strain of the vaccine virus (VR2332). Remarkably, five mutations that had independently occurred in all three vaccine-derived field isolates were discovered, pointing to strong parallel selective pressure on these positions when the vaccine virus was used in swine herds. Two of these parallel mutations were direct reversions to the parental VR2332 sequence, and were located within a papain-like cysteine protease domain and in the helicase domain [137,138]. Another critical issue is the vaccination of pregnant gilts in most pig farms. Although this practice aids in preventing PRRSV infection, it carries the risk of sow inoculation, potentially leading to fetal infection [139]. The virus can effectively cross the placenta, particularly during late-gestation stages [140]. Another study evaluated four commercial MLV vaccines by vaccinating pregnant gilts on their 100th day of gestation. Despite the vaccine strains appearing safe for the gilts, 10% of the piglets exhibited congenital infections and viremia. A phylogenetic analysis demonstrated a high similarity to the vaccine strains. Compared to non-vaccinated sows, the litters from the vaccinated sows had higher rates of weak-born piglets, mummies, and piglets with splay-leg and respiratory symptoms [141]. A more recent study corroborated these findings, detecting a similarity of over 98% between the virus and vaccine strains in piglets and gilts. In newborn piglets, cases of moderate to severe interstitial pneumonia with thickened alveolar septa were observed, along with type II pneumocyte hyperplasia. Other trial piglets showed unremarkable lung lesions. These results highlight several potential risks associated with PRRS MLV vaccination during late gestation [142].

### 4.3. KV Vaccines: A Delicate Balance between Safety and Effectiveness

The market offers limited options for KV vaccines against PRRSV (Table 3). The clear advantage of KV vaccines over MLV vaccines lies in their safety profile; their inactivated nature precludes replication, mutation, or spread within vaccinated animals. While PRRS KV vaccines are licensed for use in European countries and other parts of the world, they are not currently in use in the United States. A KV vaccine was once marketed in the U.S. under the trade name PRRomiSe, but the manufacturer discontinued it in 2005 [87]. According to prior studies, KV vaccines are primarily beneficial for their safety, but they do not consistently offer effective protection against wild strains. Assays conducted on inactivated PRRSV vaccines, with varying virus titers or different inactivation protocols, have shown no significant difference in their capacity to stimulate humoral immunity in pigs, even after three rounds of inoculation [143]. This suggests that KV vaccines are limited in their ability to induce robust humoral immune responses and control field PRRSV infections, especially against heterologous viruses. Despite eliciting a spontaneous IFN-γ response and post-challenge titers of virus-neutralizing antibodies, KV vaccines fall short in providing measurable protective immunity [144]. Moreover, in a study where two types of KV vaccines were used as a booster in conjunction with an MLV vaccine, no difference was observed in the levels of anti-N protein and anti-GP antibodies compared to those in the group receiving only the MLV vaccine. The KV vaccines also failed to enhance protection against clinical signs, virus shedding, and gross lesions. Still, there was no discernable difference in efficacy between a single injection with the MLV vaccine and a regimen combining the MLV with the KV boost vaccination [145]. Nevertheless, another study has shown that autogenous inactivated vaccines given to pregnant sows offer partial protection against PRRSV-2 to their piglets, resulting in reduced lung pathology and improved weight gain, which underscores the critical role of transferring neutralizing antibodies to piglets [146].

While KV vaccines currently play a minimal role in PRRSV protection and have seen a reduction in focus within the scientific community, they nonetheless hold potential. A recent U.S. study investigated the impact of a KV vaccine on gut microbiome diversity in pigs. The vaccine’s protective effect led to a restoration of microbiome diversity within the pigs’ gut. The study further explored microbial signatures that correlate with vaccine efficacy and immunogenicity [147]. Another study, conducted over an 18-month period, demonstrated that long-term vaccination of the female breeding stock in a farm endemic with PRRS can significantly reduce the culling rate due to reproductive failure and enhance sow longevity. These specific vaccination effects carry considerable economic implications for pork producers, particularly in closed, single-site farrow-to-finish farms [90]. Another noteworthy study found that gilts, upon receiving a combination of an MLV vaccine and autogenous inactivated vaccines (AIVs), produced neutralizing antibodies. These antibodies were then transferred to their piglets, indicating that maternal AIV boost vaccinations could enhance the protection of pre-weaning piglets against a farm-prevalent PRRSV strain [146].

### 4.4. PRRS Vaccination Programs and the Efficacy of Mixed-Antigen Vaccines in Pig Farms

A proper vaccination plan is essential for protecting swine farms from the entry and spread of diseases. Such a plan not only enhances profitability by reducing pig disease costs but also ensures the well-being of the animals and decreases the risk of zoonotic disease transmission. Currently, there is no universal vaccine strategy; vaccination plans are tailored based on geographical areas, pig flow patterns, the entry of new stock, and vaccine selection and dosing regimens, while often being dependent on pig age and category [148]. For instance, gilts require vaccinations for influenza, porcine circovirus type 2 (PCV2), and Mycoplasma (Myco) every six months before pregnancy. They also need vaccines for porcine parvovirus, Leptospirosis, and swine erysipelas about one month before breeding, and for *E. coli*, Clostridium, and Pasteurella 4–5 weeks before farrowing [20,149]. For the PRRSV MLV vaccine, it is recommended to vaccinate the entire herd twice if there is a high risk of an outbreak, such as when introducing new swine that may carry PRRSV. Additionally, gilts should be vaccinated before entering the breeding herd, and piglets should be vaccinated at the weaning stage. Vaccine usage should also align with the manufacturers’ requirements: for example, Ingelvac PRRS MLV is approved for pigs at least 3 weeks old and should not be used on boars [129,150].

PRRSV primarily infects the respiratory system in swine, while PCV2 and *Mycoplasma hyopneumoniae* are key pathogens in the porcine respiratory disease complex. In farms with inadequate disease control, co-infections are common, and their synergistic effects can severely damage the respiratory system, leading to lung lesions in dually infected pigs. Research has shown that co-infection with PCV2 and PRRSV exacerbates PCV2-associated lesions, although PCV2 does not impact PRRSV replication or lesions. Concurrent vaccination with PRRSV and PCV2 significantly reduces PRRSV viremia levels compared to single-PRRSV challenges [44,151]. Regarding co-infection with PRRSV and *M. hyopneumoniae*, it has been observed that *M. hyopneumoniae* exacerbates PRRSV-induced pneumonia, while PRRSV does not worsen pneumonia caused by *M. hyopneumoniae*. Research indicates that early vaccination with an *M. hyopneumoniae* vaccine should be prioritized to control both *M. hyopneumoniae* and PRRSV infections, especially in cases of early *M. hyopneumoniae* infection. Additionally, combined vaccination against both pathogens has been shown to effectively enhance the efficacy of individual PRRSV and *M. hyopneumoniae* vaccines, making it a promising strategy for controlling the porcine respiratory disease complex in swine production [152,153].

The phenomenon of co-infection has highlighted the urgent need for bivalent or trivalent vaccines to combat PCV2, *M. hyopneumoniae*, and PRRSV simultaneously. These pathogens are significant contributors to pneumonia in swine, collectively termed “Porcine Respiratory Disease Complex (PRDC)” [154]. In one study [155], Xu et al. utilized Baculovirus as a vaccine vector, incorporating the Cap protein from PCV2 and GP5 from PRRSV, both major immunogenic proteins that induce neutralizing antibodies and protective immunity. Fourteen days after primary immunization with this bivalent vaccine, ELISA testing detected GP5- and Cap-specific antibody responses. Following a booster immunization, the levels of GP5- and Cap-specific antibodies increased rapidly to 1:19.2 and 1:17.6, respectively. Additionally, GP5- and Cap-specific cellular immune responses demonstrated that the genetically recombinant baculovirus BacSC-Dual-GP5–Cap induced a notable cellular immune response in the swine, with a stimulation index value of 3.54 ± 0.37 [155]. Another study explored using PCV2 itself as a vector [156]. Four different linear B-cell antigenic epitopes of PRRSV were inserted into the C-terminus of the capsid gene of the PCV1-2a vaccine virus. Although this approach elicited dual humoral immune responses against both PCV2 and PRRSV in vivo, PRRSV replication was not detected in inoculated pigs, as there was no viremia or detectable viral DNA in their tissues. This suggests that either a short or insufficient antigen exposure failed to induce a sufficient immune response, or the two chimeric viruses were unable to effectively infect pigs [156]. A year later, Hu et al. employed virus-like particles (VLPs) by inserting the GP5 epitope B from PRRSV into the PCV2 Cap protein region. This modification allowed the epitope to be presented on the surface of the PCV2 VLPs, which induced strong humoral and cellular immune responses in mice, including neutralizing antibodies against both PCV2 and PRRSV. This proof of concept offers a new platform for developing bivalent or multivalent vaccines [157]. Currently, no recombinant trivalent vaccine is available to protect swine by eliciting specific antibodies against PCV2, *M. hyopneumoniae*, and PRRSV simultaneously. However, one study attempted to use a trivalent vaccine mixture to assess the synergies and effectiveness of combined vaccines. Although pigs treated with the trivalent mixture demonstrated significantly better growth performance compared to unvaccinated and challenged pigs, it did not provide equal protection compared to each respective monovalent vaccine, with the most significant disparity being observed in the response to PRRSV. This discrepancy may be attributed to the high mutability of the PRRS virus [158].

In addition to developing polyvalent vaccines for respiratory diseases, there have been various efforts to combine PRRSV vaccines with other common pig farm viruses. For example, a study by Jiang et al. utilized a live attenuated pseudorabies virus (PRV) as a vaccine vector to express the major membrane-associated proteins (GP5 or M) of PRRSV. The results showed an effective induction of PRRSV-specific immune responses and PRV-specific protective immunity, indicating its potential as a bivalent vaccine against both PRV and PRRSV infections [159]. Another study by Gao et al. involved using a PRRSV HuN4-F112 strain expressing the classical swine fever (CSF) virus E2 protein between ORF1b and ORF2a. Serological tests demonstrated that 28 days post vaccination (dpv), the piglets reached a peak level of CSFV E2 antibodies. By 42 dpv, PRRSV-specific antibodies also peaked and maintained a high level of 1:1000 until 63 dpv. Compared to the positive control group, the bivalent vaccine group exhibited high levels of CSFV-neutralizing antibodies, effectively preventing viremia [160]. Furthermore, the following animal study showed that a recombinant bivalent live vectored vaccine against CSF and HP-PRRS significantly alleviated clinical signs, yielded high antibody levels, and provided substantial protection against a challenge with a heterologous circulating NADC30-like strain. These findings underscore the potential of combining PRRSV vaccines with other virus vaccines to enhance immune protection and control multiple infections simultaneously [161].

## 5. Prospective Consideration in PRRSV Control and Vaccine Development

To date, researchers have utilized a multitude of strategies to enhance the effectiveness of MLVs and KVs against PRRSV, while also investigating alternative approaches. These methods encompass manipulating the virus to alter its antigenicity, infectivity, and pathway; integrating it with various substances such as adjuvants to reduce the effective dose or boost the immune response; and stimulating antigen-presenting cells and the recruitment of specific immune cells.

### 5.1. Advances in Viral Engineering for PRRSV Vaccine Innovation

Recent studies on the molecular virology of PRRSV and closely related nidoviruses may spark novel PRRSV vaccine design. For instance, through the use of reverse genetic systems, it has been discovered that the Serine78 site on the N protein considerably impairs the protein’s function and the replication capability of PRRSV. This site mutation has shown to significant reduce the in vitro replication efficiency of PRRSV, which potentially divulges a novel target for future vaccine development [162]. Another study demonstrated that a platform of chimeric hypo-glycosylated virus could produce a vaccine backbone, expressing the GP5 ectodomain region of a wildtype PRRSV strain with high titers, making it a viable method for the production of new vaccines to provoke an increased immune reaction against emerging PRRSV variants [163]. The advanced method of Codon Pair Deoptimization (CPD) is also garnering attention for its ability to attenuate a virus, thereby mitigating the shortcomings of MLV vaccines regarding virulence reversion, for example. This method has demonstrated effectiveness in a variety of virus vaccine models [164,165]. A South Korean study applied CPD to introduce silent mutations at the end of the infectious clone, without altering the original amino acid sequences. This intervention resulted in an increase in PRRSV-specific serum antibody levels and IFN-γ secreting cells two weeks post inoculation, and a notable reduction in gross pathological lesions in the vaccinated subjects [165]. Several European research groups have introduced a technique known as “infectious subgenomic amplicons (ISA)” utilizing the reverse genetics method. This approach involves flanking the beginning and stop positions of the viral genome with a promoter for DNA-dependent RNA polymerases, a ribozyme sequence, and a signal sequence for RNA polyadenylation. The genome is then divided into fragments, each with a region of approximately 100 base pairs overlapping with the next, to encompass the entire viral genome. This method allows for the rescue of the virus while reducing its infectivity and replication kinetics within several days [166]. Considered a promising strategy for vaccine development, this technique has already been applied to various viruses, including Zika, SARS-CoV-2, and PRRSV, showcasing its potential for broad applicability in virology research and vaccine development [167,168,169]. Additionally, the use of viral vector vaccines has been proposed to enhance cross-virus protection and broaden the immune spectrum of current PRRSV vaccines, which generally exert protection against homologous virus strains. Research on a herpesvirus-based vector embedding the Nsp5 T-cell antigen has shown that it can induce strong IFN-γ responses in the lungs and spleen and reduce lung pathology, suggesting a promising vaccine development avenue that elicits broad-spectrum T-cell and antibody responses [170]. Another study highlights the promise of mRNA vaccine development due to their high effectiveness, swift production capabilities, and potential for economical manufacturing and safe administration. Tests on animals have shown strong immunity against infectious diseases such as influenza, Zika, and rabies viruses, among others [171,172,173]. Especially during the COVID-19 pandemic, the global use of mRNA vaccines demonstrated their advantages [174]. These vaccines are non-infectious, and the mRNA is broken down through normal cellular processes, which enhances their safety. Various modifications can also enhance mRNA’s stability and translation efficiency [175]. Effective in vivo delivery can be achieved by encapsulating mRNA in carrier molecules, which facilitates rapid cellular uptake and expression [176]. In a recent study [177], Zhou et al. utilized a non-replicating mRNA vaccine expression platform to express the structural protein GP5 from a Chinese HP-PRRSV. Although swine studies were not conducted, mice studies and molecular results demonstrated that the GP5-mRNA induced high levels of GP5 antibodies. It also promoted the increased secretion of IFN-γ, TNF-α, and IL-4 by CD8+ T-lymphocytes, resulting in a stronger cellular immune response compared to the attenuated live PRRSV vaccine [177].

Studies have also put forward changes in the vaccine administration route, moving from traditional intramuscular injections using needles to needle-free devices [178,179]. This change addresses the drawbacks of the predominant pathway in pig vaccination, which includes risks like accidental needle-stick injuries and the potential for excessively strong immune responses due to the direct administration method. These needle-free devices, which have been in use for delivering antigens into the skin in human medicine, also show promising results in pigs, due to the inter-species similarity of the anatomic structure of their cutaneous system [178]. Specifically, pigs vaccinated with ID Prime Pac PRRS via the needle-free device exhibited a delayed antibody response and elevated levels of PRRSV-specific IFN-γ-secreting cells, compared to those vaccinated with Ingelvac PRRS MLV using traditional pathways and those receiving intramuscular Prime Pac vaccinations. This change in the vaccine administration pathway underscores the importance of exploring innovative methods to enhance vaccine safety and efficacy beyond novel vaccine development [179].

### 5.2. Boosting Immunity with Emerging Adjuvants

Currently, an increasing number of adjuvants, such as PCEP, Emulsigen, Carbopol, IL-6, IFN-γ, etc., are being experimentally tested in swine vaccines [180,181,182,183]. These adjuvants have demonstrated efficiency in enhancing the immunogenicity and protective efficacy of swine vaccines. More specifically, they can induce (i) short- and long-term immunity, (ii) innate and adaptive (CTL, Th1/Th2, and Ab) immune responses, and (iii) systemic and mucosal immunity. They are also considered to be relatively stable, safe, and cost-effective for pharmaceutical development. In a recent study, researchers employed a novel PRRSV-specific IgM monoclonal antibody (Mab)–PR5nf1 (Mab-PR5nf1) as an innovative adjuvant. This was used to formulate a cocktail composed of inactivated PRRSV and Mab-PR5nf1, alongside a standard Montanide™ ISA 206 adjuvant, aimed at enhancing the efficacy of the PRRSV-KV vaccine. The findings suggested a substantial improvement in the overall survival rate and cell-mediated immunity. Additionally, viral shedding was reduced and serum neutralizing antibody levels were elevated [184]. Another kind of study highlighted the remarkable antiviral properties of IFNs, underscoring their potential to amplify immune responses in pigs and provide a crucial layer of protection [84]. Through meticulous in vitro and liver organ studies, researchers discovered that the IFN-α, IFN-β, and IFN-ω subtypes offer robust immunity and superior protection at the molecular level [82,83]. This leads to reduced viral-induced stress on organs and the immune system. Additionally, a notable under-expression of genes associated with acute inflammatory responses was observed. These findings suggest that employing IFN as a co-expressed adjuvant could be a strategic means to counteract PRRSV’s immune suppression and maintain optimal cellular homeostasis [67,82,83,185].

Several studies have also highlighted Toll-like receptor (TLR) agonists as effective adjuvants [186,187]. These are known to fortify Th1-driven immune responses and stimulate antigen-presenting cells (APCs), thereby improving humoral immunity in both neonatal and adult pigs [24,187]. A recent study demonstrated that combining a MLV PRRS vaccine (Ingelvac^®^ PRRS) with a water-soluble whole-cell lysate (WCL) from heat-killed *Mycobacterium tuberculosis* (*Mtb*) supports body weight gain in infected pigs, diminishes lung pathology, and enhances PRRSV-neutralizing antibody levels while decreasing viremia [138]. Analyses have suggested that the adjuvant action of *Mtb* WCL, potentially through a regulatory release of IL-6, may contribute to triggering a PRRSV-specific cellular-mediated immune response [188]. In a separate investigation, porcine CpG oligodeoxynucleotides (pCpG ODN) were tested as an adjuvant with a PRRSV KV vaccine to evaluate their impact on immune responses. The findings showed that a low dosage (500 μg) of CpG improved immune responses, despite the absence of any statistically significant differences in the antibody ELISA titers compared to the vaccine-only group. However, the low-dose vaccine pCpG ODN-treated groups showed more severe clinical symptoms, such as those related to body temperature, weight gain, and overall behavior, than the full-dose vaccine-only group and the group given the full-dose vaccine plus pCpG ODN [189]. Both of the above studies utilizing either MLV or KV vaccines with distinct TLR-agonist adjuvants demonstrate their potential in enhancing the immune defense of pigs against PRRSV infections, albeit with obstacles that remain in need of further investigation. Other PRR-related adjuvants (e.g., STING agonists) showing potential in SARS-CoV-2 studies have yet to be explored for PRRSV, despite both viruses belonging to the *Nidovirales* order [190]. These STING agonists activate the STING signaling pathway to induce an IFN-I response and the production of pro-inflammatory cytokines. Research has pinpointed the effectiveness of the STING agonist CF501 as an adjuvant in conjunction with the SARS-CoV-2 RBD-F subunit vaccine, which stimulated an enhanced cellular immune response in both mice and rabbits [140]. Additionally, rhesus macaques receiving the CF501/RBD-Fc vaccine showed durable immunity with a high SARS-CoV-2 RBD-specific IgG level, nearly 18 times greater than that achieved with the control alum/RBD-Fc groups [190]. Another study employed the nanoscale adjuvant zeolitic imidazolate framework-8 (ZIF-8), a TLR agonist, which fine-tuned immune reactions to the SARS-CoV-2 RBD trimer at both cellular and molecular levels. Mice vaccinated with GR-ZIF displayed an eight-fold surge in IFN-γ levels compared with PBS-treated controls and a significant increase in the expression of 55 genes linked to innate, B-cell, and T-cell functions. These insights are crucial for advancing PRRSV adjuvant vaccine research [191].

Future studies on these adjuvants could provide alternative means to enhance the protective efficiency of KV vaccines, stimulate mucosal immune responses, and improve the safety of MLV vaccines by minimizing viral shedding and reducing the possibility of reversion to virulence [192].

### 5.3. Cross-Virus Insights for Future PRRSV Vaccine Development

Research into diverse viruses has ushered in innovative approaches to PRRS vaccination. A key area of interest is the development of nanoparticle subunit vaccines [193,194,195]. The inclusion of nanoparticles in vaccine compositions enhances antigen stability and immunogenicity, offers precise delivery, and allows for a gradual antigen release. This is possible because their size is comparable to that of subcellular particulate components, enabling them to enter cells via endocytosis, especially pinocytosis (Figure 1). Various types of nanoparticles have been used, including polymeric nanoparticles, inorganic nanoparticles, liposomes, virus-like particles, self-assembling proteins, and oil–water emulsions [193]. By chemically linking protein antigens to carrier molecules, forming nanoparticles, this method bolsters immunogenicity, minimizes antigen breakdown, and promotes a broader immune reaction [194,196]. A recent investigation revealed that nanoparticles made from polylactic acid (PLA) carrying inactivated PRRSV combined with heat-labile enterotoxin subunit B (LTB) served as an effective mucoadhesive vaccine delivery system. This formulation not only triggered systemic and mucosal immunity against PRRSV but also led to a marked rise in IFNγ-producing cells and a significantly higher neutralizing antibody titer compared to control groups. And the levels of both IgG and IgA antibodies in the pigs receiving the adjuvant-enhanced nanoparticles continued to rise, registering significantly higher measures than those of other groups over 35 days post immunization [195]. Moreover, an exploration into iron-based antiviral agents, specifically gelatin-stabilized ferrous sulfide nanoparticles (Gel-FeS NPs) targeting PRRSV, has demonstrated their virucidal and antiviral capabilities, suggesting a novel approach for employing iron-based nanoparticles in acting against PRRSV [197].

Another approach involves a subcutaneous vaccine that uses a live Zika virus encapsulated within a self-adjuvanting hydrogel [198]. This technology has emerged as the pig industry seeks more effective and safer vaccination methods to enhance animal welfare. Extensive studies have demonstrated that the intradermal (ID) delivery system can provide protection comparable to or better than that provided by the intramuscular (IM) route [179,199,200,201]. This method is effective for PRRS vaccination, as the skin contains cells from both the innate and adaptive immune systems. This presence may explain why skin-based vaccination elicits stronger systemic immune responses [202]. Potent responses could enable vaccine dose-sparing, as demonstrated with influenza and rabies vaccines [203,204]. One study utilizing Ingelvac PRRS MLV and Prime Pac PRRS MLV involved vaccinating 3-week-old castrated male pigs. Following a virus challenge, the IM vaccination group exhibited a higher PRRSV-specific antibody response according to ELISA testing, whereas the ID vaccination group showed the highest expression levels of PRRSV-specific IFN-γ-SC 21 days post vaccination, indicating a stronger cell-mediated immune response [179]. Additional studies have also shown that ID vaccination of suckling piglets positively affects their health and performance, inducing an immune response equivalent to that achieved with the traditional intramuscular route [199,200,201].

Subcutaneous vaccines employ chitosan scaffolds, which, due to their positively charged side chains, can capture viruses through electrostatic interactions. These scaffolds not only facilitate virus entrapment using wildtype field strains but also stimulate innate immune responses and attract immune cells via the intradermal activation of PRRs [198,205]. Hydrogels are particularly advantageous for their capacity to encapsulate a wide range of living viruses, viral antigens, and adjuvanting immunomodulators, due to their efficient encapsulation capabilities [206]. By enabling the controlled release of these substances, hydrogels effectively prime the recruitment of immune cells directly at the cutaneous site of administration, which, in turn, enhances the vaccine’s effectiveness directly using natural virus strains or intact viral antigens. This solution may effectively address several vaccination challenges, including precise vaccine delivery to the target site, the prolonged release of vaccine molecules, and the handling of chemically diverse antigen molecules [205,207]. According to the conceptual experiment (Figure 1B), viral particles of PRRSV would be mixed with chitosan oligosaccharide (COS) of an opposite charge, along with sodium bicarbonate (NaHCO3) and calcium chloride (CaCl2), to create the virus-encapsulating hydrogel. This gel then forms an inflammatory microenvironment that is ideal for the absorption and processing of the targeted virus, wherein PRRs are triggered, leading to the recruitment of immune cells and the initiation of innate immune mechanisms to mimic natural infections [208]. This hydrogel/nanoparticle platform thus enables the direct use of the field virus isolate to induce effective antigen presentation, an increase in germinal center (GC) B-cells, and enhanced cross-presentation within lymph nodes. If validated, this strategy may be feasible to stimulate immune cells using a virulent field PRRSV isolate trapped locally under the skin to induce a systemic immune response, thus effectively protecting swine from a deadly viral threat [198,208]. In a nutshell, evolving PRRS strains poses a challenge for vaccine development, and recent strategic studies of other viruses and novel adjuvant platforms offer valuable insights and promises for designing more effective vaccines against PRRS.

## 6. Conclusions

PRRS, caused by the highly mutable positive-sense single-strand-RNA PRRS virus, continues to result in significant economic losses within the swine industry due to respiratory illnesses, reproductive failure, and stunted growth. Despite three decades of research and the development of some partially protective vaccines, a truly effective solution remains elusive. Both MLV and KV vaccines play essential roles in PRRS control. MLV vaccines are noted for their strong immune responses and longer duration of immunity but carry potential risks, particularly when used in pregnant gilts. Conversely, KV vaccines offer a safer profile but provide weaker and shorter-lived immunity. The complex immunological interactions and the ever-evolving nature of PRRSV present significant challenges to vaccine development. Limited knowledge of the virus’s interaction with host cells, fusion mechanisms, IFN antagonism, and immune evasion strategies further hinder the development of more effective vaccines. As this virus continues to evolve, future vaccines must ensure safety and provide protection against both known and emerging strains.

Continued research and innovation are essential to achieve next-generation vaccines that offer robust protection with minimal risks. Integrating these advanced vaccines into comprehensive farm health programs is vital for the sustainable management of PRRS and the improvement of swine health and productivity.

## Figures and Tables

**Figure 1 vaccines-12-00606-f001:**
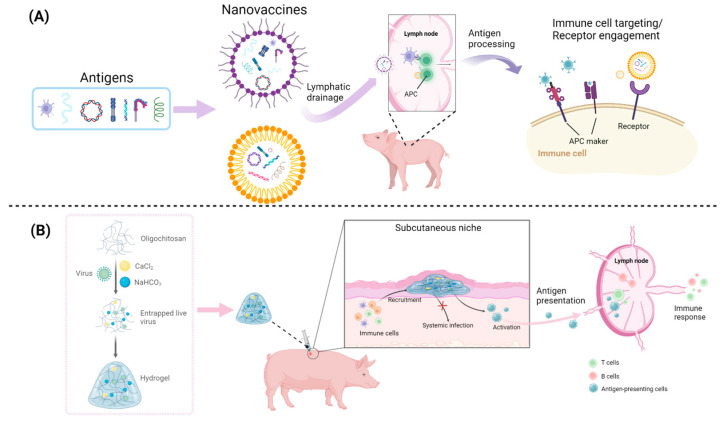
Vaccine hypotheses—nanoparticle and self-adjuvanting hydrogel vaccine design strategies. (**A**) Nanoparticle vaccines utilize a versatile array of carriers—like liposomes, emulsions, dendrimers, and nanogels—to encapsulate antigens and adjuvants. These nanovaccines are designed for optimal lymph-node delivery via the lymphatic system, safeguarding the payload from degradation. Upon arrival, they release their contents to antigen-presenting cells (APCs) for effective immune activation. Additionally, these nanoparticles can be engineered to target specific immune cell subsets and intracellular pathways, enhancing the immune response precision. (**B**) This subcutaneous PRRSV vaccine encapsulated in a self-adjuvanting hydrogel, upon administration, activates PRRs, recruits immune cells, and triggers innate immunity. This sequence fosters lymph-node antigen presentation and GC B-cell proliferation, leading to a strong, specific immune response and memory against PRRSV.

**Table 1 vaccines-12-00606-t001:** The advantages and disadvantages of KV and MLV vaccines [86,87,88,89,90].

Aspect	KV Vaccines	MLV Vaccines
Immune Response	Weaker immune response	Stronger immune response
Safety	Safer, as the virus is inactivated	Higher risk, as the virus is live but attenuated
Duration of Immunity	Shorter duration	Longer duration
Onset of Immunity	Slower onset	Faster onset
Vaccine Stability	More stable and less sensitive to storage conditions	Less stable and more sensitive to storage conditions
Cost	Generally cheaper	Generally more expensive
Reversion to Virulence	No risk of reversion to virulence	Potential risk of reversion to virulence
Use in Pregnant Animals	Safer for use in pregnant animals	Riskier for use in pregnant animals
Efficacy against Diverse Strains	Less effective against diverse viral strains	More effective against diverse viral strains

**Table 2 vaccines-12-00606-t002:** List of PRRSV commercial MLV vaccines [91,92,93,94,95,96,97,98,99,100,101,102,103,104,105,106,107,108,109,110,111,112,113,114,115,116,117,118,119,120,121,122,123,124].

MLV Vaccine Name	Parental Strain	Type	Usage	Introduction to the Market	Predominant Region(s) Using the Vaccine	Developer/Company
Ingelvac PRRS MLV	VR-2332	PRRSV-2	Sow, Piglet	1994	Africa; Asia; Europe; North America; South America	Boehringer Ingelheim
Ingelvac PRRS ATP	JA-142	PRRSV-2	Sow, Piglet	2004	Asia; Europe; North America	Boehringer Ingelheim
Fostera PRRS	P129	PRRSV-2	Sow, Piglet	2012	Africa; Asia; Europe; North America	Zoetis
Prime Pac PRRS	Neb-1	PRRSV-2	Sow, Piglet	2014	Africa; Asia; Europe; North America	Merck Sharp & Dohme
Prevacent PRRS	RFLP 184	PRRSV-2	Sow, Piglet	2018	Asia; Europe; North America	Elanco
Porcilis PRRS	DV	PRRSV-1	Sow, Piglet	2000	Asia; Europe	Merck Sharp & Dohme
Unistrain (Amervac-) PRRS	VP-046 BIS	PRRSV-1	Sow, Piglet	2013	Africa; Asia; Europe	Hipra
ReproCyc PRRS EU	94881	PRRSV-1	Sow	2015	Africa; Asia; Europe	Boehringer Ingelheim
Ingelvac PRRSFLEX EU	94881	PRRSV-1	Piglet	2015	Africa; Asia; Europe	Boehringer Ingelheim
Pyrsvac-183	ALL-183	PRRSV-1	Sow	1999	Asia; Europe	Syva
Suvaxyn PRRS MLV	96V198	PRRSV-1	Sow, Piglet	2017	Europe	Zoetis
R98	R98	PRRSV-2	Sow, Piglet	2011	Asia	Tianjin Ringpu
GDr180	GDr180	PRRSV-2	Sow, Piglet	2015	Asia	China Institute of Veterinary Drug Control
CH-1R	CH-1R	PRRSV-2	Sow, Piglet	2007	Asia	Harbin Veterinary Research Institute
HuN4-F112	HuN4-F112	PRRSV-2	Sow	2011	Asia	Harbin Veterinary Research Institute
TJM-F92	TJM-F92	PRRSV-2	Sow	2013	Asia	Institute of Special Animal and Plant Sciences
JXA1-R	JXA1	PRRSV-2	Sow, Piglet	2008		Chinese Center for Animal Disease Control and Prevention

**Table 3 vaccines-12-00606-t003:** List of PRRSV commercial KV vaccines [143,144,145,146,147].

KV Vaccine Name	Parental Strain	Type	Usage	Introduction to the Market	Predominant Region(s) Using the Vaccine	Developer/Company
Suipravac-PRRS	VP-046 BIS	PRRSV-1	Sow	N/A	Europe	Hipra, Girona, Spain
Progressis	P120	PRRSV-1	Sow	N/A	Asia; Europe	Ceva Santé Animale, Mayenne, France
SUIVAC PRRS-Ine/SUIVAC PRRS-In	VD-E1/VD-E2/VD-A1	PRRSV-1	Sow, Piglet	2017 *	Europe	Dyntec, Terezín Czech Republic

Note: * The accuracy of data on the year of introduction is limited by the availability of online information and has been verified based on information disclosed by the European Medicines Agency. N/A, not available.

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
