# Peer review of "Current Status of Vaccines for Porcine Reproductive and Respiratory Syndrome: Interferon Response, Immunological Overview, and Future Prospects"

_vaccines, 2024, doi:10.3390/vaccines12060606_

Round 1

Reviewer 1 Report

Comments and Suggestions for Authors

Major comments

-4.2. Potential Risks and Challenges of MLV Vaccines: you should also report data from more recent studies about the risk of administration of MLV during the last stage of gestation (>95th day of gestation) due to the ability of the vaccine virus to cross the placenta (e.g. negative effects on embryos and neonatal piglets, such as an increase of mummies and stillbirth piglets, lung lesions in piglets)

-5.3. Cross-Virus Insights for Future PRRSV Vaccine Development: you could add a couple of paragraphs about recent data regarding the advantages and disadvantages and the efficacy of intradermal vs intramuscular vaccination against PRRSV. For example, animal welfare issues are related to the method of the applied vaccination. 

 -You could also report a few data about the current bivalent PRRSV/PCV2 vaccine or trivalent vaccine mixture against a triple challenge with Mycoplasma hyopneumoniae, PCV2, and PRRSV

·        

Minor comments

- You could summarize the advantages and disadvantages of KV and MLV vaccines in a table, including appropriate references.

Author Response

Enclosed, please find our revision of the manuscript entitled: “Current Status of PRRS Vaccines: Interferon Response, Immunological Overview and Future Prospects” that we are submitting for consideration in Vaccines per the invitation of the Guest Editor of the special issue  “Interferon Responses after Vaccine Administration”.

Below, we provide detailed point-by-point answers addressing the reviewers’ comments with the revised parts in red text in the proof attached. We would like to extend our thanks to the reviewers for their positive feedback and constructive comments. We believe that the revised version is a better fit for the journal and the special issue.

Reviewer 2 Report

Comments and Suggestions for Authors

The manuscript provides a review regarding PRRS vaccinations of pigs.

Can the authors please provide a list with the other recent reviews on the subject and their time of publication? This will help to make sure that there is a need for a relevant review.

Also, the authors must underline the differences between the present manuscript and previous reviews, in order to ensure that there are novel information and details presented in there.

The number of references is very satisfactory. Nevertheless, the authors should present in brief their methodology for including those references. I understand that this is a narrative review, but still the authors had a strategy in selecting references for inclusion and this strategy must be described to some extent.

After mentioning the previous reviews on the same topic, please try to focus on including more references published subsequently to the previous reviews.

Please include a new section to describe the use of PRRS vaccines within the frame of vaccination programs in pig farms, i.e., when to vaccinate against PRRS in conjunction with other vaccinations scheduled in pig farms.

Also, please add a comment regarding mixed antigen vaccines, i.e., against PRRS and against other pig infections.

Finally, the conclusions are not fully in line with the text, so please rewrite by toning down.

Author Response

(The authors gave the same response as above.)

Round 2

Reviewer 2 Report

Comments and Suggestions for Authors

All issues have been corrected. No more comments.